# Targeted Sphingolipid Analysis in Heart, Gizzard, and Breast Muscle in Chickens Reveals Possible New Target Organs of Fumonisins

**DOI:** 10.3390/toxins14120828

**Published:** 2022-11-24

**Authors:** Philippe Guerre, Caroline Gilleron, Maria Matard-Mann, Pi Nyvall Collén

**Affiliations:** 1National Veterinary School of Toulouse, ENVT, Université de Toulouse, F-31076 Toulouse, France; 2Olmix S.A., ZA du Haut du Bois, F-56580 Bréhan, France

**Keywords:** fumonisin, sphingolipid, ceramide, sphingomyelin, muscle, heart

## Abstract

Alteration of sphingolipid synthesis is a key event in fumonisins toxicity, but only limited data have been reported regarding the effects of fumonisins on the sphingolipidome. Recent studies in chickens found that the changes in sphingolipids in liver, kidney, lung, and brain differed greatly. This study aimed to determine the effects of fumonisins on sphingolipids in heart, gizzard, and breast muscle in chickens fed 20.8 mg FB1 + FB2/kg for 9 days. A significant increase in the sphinganine:sphingosine ratio due to an increase in sphinganine was observed in heart and gizzard. Dihydroceramides and ceramides increased in the hearts of chickens fed fumonisins, but decreased in the gizzard. The dihydrosphingomyelin, sphingomyelin, and glycosylceramide concentrations paralleled those of ceramides, although the effects were less pronounced. In the heart, sphingolipids with fatty acid chain lengths of 20 to 26 carbons were more affected than those with 14–16 carbons; this difference was not observed in the gizzard. Partial least squares-discriminant analysis on sphingolipids in the heart allowed chickens to be divided into two distinct groups according to their diet. The same was the case for the gizzard. Pearson coefficients of correlation among all the sphingolipids assayed revealed strong positive correlations in the hearts of chickens fed fumonisins compared to chickens fed a control diet, as well as compared to gizzard, irrespective of the diet fed. By contrast, no effect of fumonisins was observed on sphingolipids in breast muscle.

## 1. Introduction

Fumonisins are mycotoxins produced by molds of the genus *Fusarium*, mainly *F. verticillioides*. This class of contaminant is found worldwide in maize, maize byproducts, and animal feed [1,2]. Among the various types of fumonisins, fumonisins B are the most abundant, and among fumonisins B, fumonisin B1 (FB1), and to a lesser extent fumonisin B2 (FB2), are the most abundant, toxic, and regulated. Fumonisins B are considered carcinogenic in rodents and probable carcinogens for humans, with a provisional maximum tolerable daily intake (PMTDI) being set for humans and maximum levels of fumonisins being recommended in animal feed [2,3,4,5]. Fumonisins have multiple toxic effects in animals as they are neurotoxic in horses, pneumotoxic in pigs, hepatotoxic, and nephrotoxic in most animal species, including poultry. At the cellular level, oxidative stress, mitochondrial dysfunction, and induction of apoptosis or cell proliferation have been reported [6,7,8,9]. Although the effects of FB1 vary greatly, inhibition of ceramide synthase (CerS) due to structural analogy between FB1 and sphingoid bases is considered one of the main mechanisms underlying the toxicity [10,11]. This inhibition leads to reduction of the de novo synthesis of ceramides and an increase in sphinganine (Sa) concentrations, while sphingosine (So) concentrations tend to decrease (Figure 1). Ceramide levels in cells can be maintained by hydrolysis of sphingomyelins, which is known as the salvage pathway for ceramide synthesis.

The pronounced effect of fumonisins on sphingolipids has led to widespread use of the Sa:So ratio as an exposure biomarker in humans and animals, including poultry [9,11,12]. There have, however, only been a limited number of studies to date aimed at characterization of the effects of fumonisins on the sphingolipidome. Sphingolipidomic studies in chickens and turkeys conducted at the maximum recommended concentration for fumonisins in feed in the European Union have revealed numerous alterations in sphingolipids in the liver [13,14]. These changes are consistent with the previously reported inhibition of CerS in vitro and in vivo [10,11]. Because the effects of fumonisins on sphingolipids in chicken and turkey livers were characterized by a decrease in C14–C16 ceramides and an increase in C20–C24 ceramides, it has been hypothesized that inhibition of CerS may be more pronounced for CerS5 than that for CerS2 (Figure 1), whereas the opposite has been suggested to occur in mammals [11,15]. This difference is important because a decrease in C20–C24 ceramides, and a compensatory increase in C16 ceramide, have been reported in CerS2 knockout mice, and these changes were responsible for hepatotoxicity that resembles fumonisins toxicity [16,17]. Sphingolipid analysis in chickens and turkeys also revealed that sphingomyelin levels did not decrease in liver [13,14]. This latter result is also important for explaining the lack of toxicity of fumonisins in these studies, with a decrease in sphingomyelins being reported in tissue and cell cultures at toxic concentrations of fumonisins, and this event was found to precede ceramide-induced apoptosis [11,15,18,19,20,21].

Targeted sphingolipids analysis conducted in the kidneys, lungs, and brains of chickens revealed that the effects of fumonisins were not limited to changes in sphingolipids in the liver, and that the effects varied according to the organ studied [22]. The effects in kidneys were similar to those observed in livers (Figure 1). By contrast, sphinganine and ceramides were unchanged in the lungs and brain, suggesting no inhibition of CerS in these organs. However, major changes in sphingolipid concentrations in lung and brain were found in chickens fed with a diet that was contaminated with fumonisins compared to controls [22]. In the lungs, these changes corresponded to an increase in sphinganine-1-phosphate and a decrease in glycosylceramides. In the brain, these changes corresponded to an increase in deoxysphinganine, sphingosine, ceramides, and sphingomyelins (Figure 1). Interestingly, the most affected sphingolipids in lung and in brain in chickens fed fumonisins corresponded to compounds that have key functions in these organs in human diseases [23,24,25,26]. As only traces of FB1 were found in lungs and no FB1 was detected in brains, it has been hypothesized that the effects of fumonisins are due to indirect effects, metabolites, or cell-mediated (Figure 1) [22].

Although fumonisins have been reported to exert cardiac toxicity in mammals and avian species [27,28,29,30], the effect of fumonisins on sphingolipids in the heart remain unknown. These effects seem interesting to study as sphingolipids have a key role in the onset of cardiovascular diseases in humans [31]. Additionally, diets containing fumonisins have been reported to damage the gizzard and to alter its weight [32,33,34,35,36], although there have been no studies to date regarding the effect of fumonisins on sphingolipids in the gizzard. Moreover, sphingolipids are known to play a key role in skeletal muscle [37,38,39,40], but the effects of fumonisins on this tissue remain unknown. The purpose of this study was to investigate the effects of 20.8 mg FB1 + FB2/kg fed to chicken for 9 days on sphingolipids in the heart, gizzard, and breast muscle. Targeted sphingolipid analysis was conducted by UHPLC-MSMS of samples obtained from animals by a method that has already been used to characterize the effects of fumonisins on sphingolipids in liver, kidney, lung, and brain [13,22].

**Figure 1 toxins-14-00828-f001:**
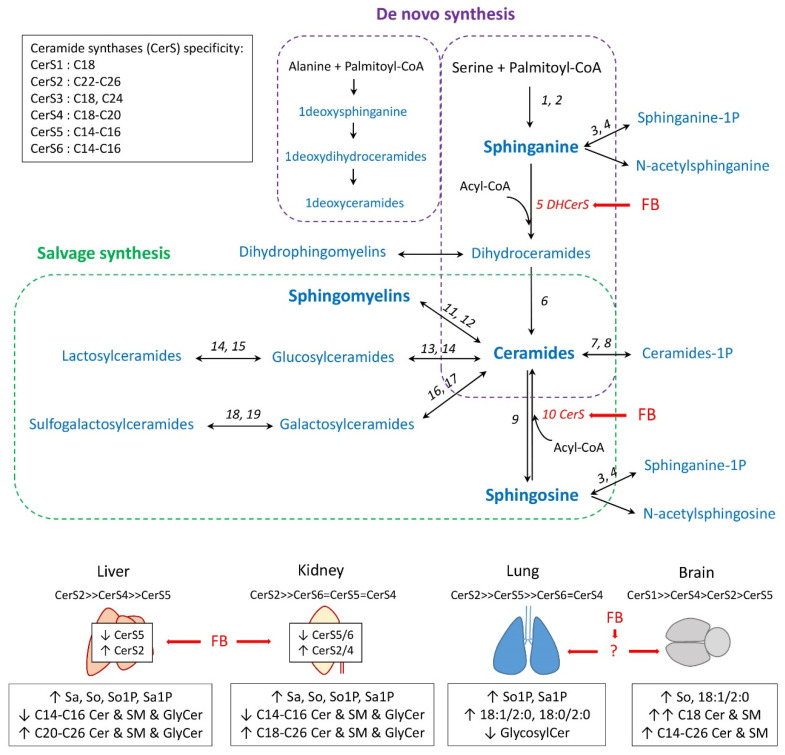
Schematic outline of the sphingolipid synthesis pathways [11]. Ceramide synthases (CerS) specificity and tissue expression were obtained from [41]. Fumonisins B (FB) are inhibitors of CerS (red arrow). 1-Deoxysphinganine is generated from alanine [42]. The effects of fumonisins on sphingolipids in liver, kidney, lung, and brain are summarized from [13,22]. 1: Serine palmitoyl-transferase; 2: Reductase; 3: Sphinganine kinase; 4: Phosphatase; 5: (Dihydro)ceramide synthase; 6: Dihydroceramide desaturase; 7: Ceramide kinase; 8: Phosphatase; 9: Ceramidase; 10: Ceramide synthase; 11: Sphingomyelin synthase; 12: Sphingomyelinase; 13: Glucosylceramide synthase; 14: β-glucosidase; 15: Lactosylceramide synthase; 16: Ceramide galactosyltransferase; 17: Galactosylase; 18: Cerebroside sulfotransferase; and 19: Arylsulfatase.

## 2. Results

### 2.1. Effect of Fumonisins According to the Class of Sphingolipids

The concentrations of sphingolipids measured in heart, gizzard, and breast muscle of chickens fed the control diet and chickens fed a diet containing 20.8 mg FB1 + FB2/kg over a period of 9 days are presented in Table 1. The effects of fumonisins according to the class of sphingolipids are presented in Figure 2.

Feeding fumonisins contaminated diet significantly increased the Sa:So ratio in heart, whereas no significant differences between the controls and the treated samples were observed in gizzard and breast muscle. The effects of fumonisins on dihydroceramides and on ceramides also varied according to the tissue studied. A significant increase in dihydroceramides and ceramides was observed in the heart, whereas a significant decrease was observed in the gizzard. No effect was observed on breast muscle (Figure 2). No significant effect of fumonisins was observed on the concentrations of dihydrosphingomyelins, sphingomyelins, or glycosylceramides, irrespective of the tissue.

### 2.2. Effects of Fumonisins on Sphingolipids in Heart

Being fed a diet containing 20.8 mg FB1 + FB2/kg over a period of 9 days significantly increased the Sa level in heart (Table 1). A significant increase in 18:1/20:0, 18:1/22.1, 18:1/22:0, 18:1/23:0, 18:1/24:2, and 18:1/26:2 ceramides was also observed. The levels of most of the other ceramides and dihydroceramides were increased by fumonisins, but the effects were not significant. SM18:1/20:0 was the only sphingomyelin significantly increased in heart by fumonisins. The effects of the toxins on other sphingomyelins varied according to the fatty acid carbon chain length. SM14–16 decreased slightly, whereas a weak but not significant increase was observed for SM18–SM26. Concerning glycosylceramides, the levels of Hex18:1/22:0, Hex18:1/24:1, Lac18:1/24:1, Lac18:1/24:0, and ceramide sulfatide ST18:1/24:0 were significantly increased by fumonisins. The levels of all other hexosylceramides and lactosylceramides were also increased, but the effects were not significant (Table 1).

Partial least square-discriminant analysis was conducted to determine whether changes in the heart sphingolipidome were sufficient to separate chickens into two groups according to their exposure to fumonisins. As shown in Figure 3C, a good separation of chickens was observed in this analysis. The values of the R^2^Y and R^2^X indices were 0.608 and 0.632, respectively, indicating that the selected sphingolipids could predict the group that the chickens belonged to. The value of Q^2^ with the two first components was 0.403, indicating that the model was of medium quality (Figure 3D). Not surprisingly, sphingolipids that were important in the projection corresponded to variables that differed significantly in chickens fed fumonisins versus the controls. It is also interesting to note that SM18:1/14:0 and SM18:1/16:0 were important in explaining the repartition of chickens into the two groups (Figure 3A,B), whereas only a very weak difference was noted in the mean concentration of these analytes in heart compared to the controls (Table 1). Other variables important in the projection corresponded to sphingolipids that were increased by fumonisins for which the changes measured were considered not significant by ANOVA.

Figure 4 shows the correlation observed in heart for So, Sa, and sphingolipids with 14 to 24 carbon fatty acid saturated chain lengths. The numerical values of the Pearson coefficients of correlation are reported in Appendix A.

Only weak correlations were observed between Sa and So, among the sphingoid bases and the other sphingolipids in control chickens (i.e., unexposed to fumonisins) (Figure 4A). Moreover, the correlations among the different classes of sphingolipids were weak, except for ceramides and glycosylceramides, which exhibited significant positive correlations, and ceramides and sphingomyelins, which exhibited significant negative correlations. By contrast, significant positive correlations were observed within the same class among dihydroceramides, ceramides, sphingomyelins, hexosylceramides, and lactosylceramides, which have similar fatty acid chain lengths. The feeding of fumonisins strongly increased the positive correlations measured among sphingolipids in the heart (Figure 4B). Notably, significant positive correlations among the different classes of sphingolipids were observed in chickens fed fumonisins, which was not the case in the controls. Moreover, the coefficients of correlation found among the same class of sphingolipids with similar fatty acid chain lengths were generally increased (Appendix A). The correlations among unsaturated sphingolipids and other analytes were close to those observed for the corresponding saturated forms. The correlations among C25 and C26 sphingolipids and other analytes were close to those observed for C24 sphingolipids (Appendix A).

All of these results suggest a significant effect of fumonisins on sphingolipids in the heart. This effect was dominated by an increase in dihydroceramides and ceramides and, to a lesser extent, an increase in glycosylceramides. This was accompanied by strong positive correlations between the different classes and subclasses of sphingolipids. The effects of fumonisins also varied according to the fatty acid chain length. C20–C26 ceramides were significantly increased by the addition of the toxins to the feed (*p* = 0.029), whereas C14–C16 ceramides were not (*p* = 0.112). Additionally, Hex20–Hex26 hexosylceramides, Lac20–Lac26 lactosylceramides, and SM20–SM26 sphingomyelins tended to increase, whereas Hex16, Lac16, and SM14SM16 were unaffected.

### 2.3. Effects of Fumonisins on Sphingolipids in Gizzard

As shown in Table 1, the sphinganine concentration in the gizzards of chickens fed a diet containing 20.8 mg FB1 + FB2/kg over a period of 9 days was significantly increased. A significant increase in N-acetylsphinganine (d18:0/2:0) was also observed. By contrast, the levels of most of the dihydroceramides and most of the ceramides assayed were decreased by fumonisins, and this effect was significant for 18:1/14:0, 18:1/16.0, 18:0/16:0, 18:0/18:0, 18:1/22:2, 18:0/22:0, 18:1/23:0, 18:1/24:2, 18:1/26:2, 18:1/26:1, and 18:1/26:2. No significant difference between groups was observed in terms of the dihydrosphingomyelin and sphingomyelin levels in gizzard. Hex18:1/24:1 and Lac18:1/16:0 were significantly decreased by fumonisins, and most of the other glycosylceramides tended to decrease, but the effect was not significant (Table 1).

Figure 5 shows the results of the partial least square-discriminant analysis conducted on the sphingolipids in gizzard. A good separation of the chickens according to the feed was observed (Figure 5C). The values of the R^2^Y, R^2^X, and Q^2^ indices, which were 0.755, 0.615, and 0.671, respectively, indicated the good predicted appurtenance of the chickens to the different groups and the good quality of the model. The confusion matrix confirmed the model was highly sensitive and specific (Figure 5D). The sphingolipids that were important in the projection mainly corresponded to the dihydroceramides, ceramides, and glycosylceramides that were significantly decreased in chickens fed fumonisins. SM18:1/16:0 and SM18:1/18:0 also had VIP scores above 1.1 in this analysis (Figure 5A,B), although their concentrations in gizzards did not differ from the controls (Table 1).

The correlations measured among the assayed sphingolipids in gizzards in this study are reported in Figure 6 and Appendix A. Strong significant positive correlations were observed among So and dihydroceramides and among So and ceramides, whereas weak negative correlations were observed among Sa and dihydroceramides in chickens unexposed to fumonisins (Figure 6A). Significant positive correlations were observed among sphingolipids of the same class with similar fatty acid chain lengths. Additionally, a significant positive correlation was observed between dihydroceramides and ceramides and between sphingomyelins and hexosylceramides. Most of the correlations between ceramides and sphingomyelins, ceramides and hexosylceramides, and between hexosylceramides and lactosylceramides were weak and not significant. Feeding fumonisins only resulted in weak effects on the correlations observed among sphingolipids (Figure 6B). The correlations among unsaturated sphingolipids and other analytes were close to those observed for the corresponding saturated forms. The correlations among C25 and C26 sphingolipids and other analytes were close to those observed for C24 sphingolipids (Appendix A).

These results suggest that there was a significant effect of fumonisins on sphingolipids in gizzards in this study. The effects observed were dominated by a decrease in dihydroceramides and ceramides. Sphingomyelins and glycosylceramides also tended to decrease in gizzards of chickens fed fumonisins, although the effects were not significant at the class level and varied with the analyte assayed. The feeding of fumonisins did not profoundly change the correlations among the sphingolipids in the gizzard.

### 2.4. Effects of Fumonisins on Sphingolipids in Breast Muscle

The concentrations of sphingolipids measured in breast muscle of chickens fed a control diet free of mycotoxins and chickens fed 9 days with a diet containing 20.8 mg FB1 + FB2/kg are reported in Table 1. Sphingosine was the only sphingolipid assayed for which the concentration in gizzard was significantly affected by fumonisins, and its concentration increased. There were only weak non-significant changes in the concentrations of dihydroceramides, ceramides, sphingomyelins, hexosylceramides, and lactosylceramides.

Partial least square-discriminant analysis was conducted to determine whether this analysis would be able to discriminate chickens according to the diet that they had been fed (Appendix A). As shown in Appendix A, there was a good separation of chickens. The values of the R^2^Y and the R^2^X indices were satisfactory, and the confusion matrix revealed the model was highly sensitive and specific (Appendix A). However, the value of Q^2^ obtained with the two first components was only 0.185, indicating that the model can vary greatly depending on the sphingolipids selected for the modeling. Additionally, the standard deviations observed for the score of the variables important in the projection (VIP) were high in breast muscle (Appendix A), confirming high variability in the modeling.

Finally, all together, the total amount of sphingolipids (Figure 2), the concentration of the various analytes assayed (Table 1), and the comparison of the repartition of the chickens into groups according to the diet that they had been fed using PLS-DA (Appendix A) revealed that the presence of fumonisins in the feed only had a minor effect on the sphingolipids in breast muscle in this study.

## 3. Discussion

Feeding 20.8 mg FB1 + FB2/kg over a period of 9 days did not lead to significant differences in the performances or the biochemistry in the chickens in this study [43]. This observation is consistent with the regulatory guidelines, which define a maximum tolerable level of fumonisins in feed of 20 mg FB1 + FB2/kg [1,4], whereas feeding 2.5 mg FB1 + FB2/kg of feed was reported to alter the length of intestinal crypts in chickens after 21 days of exposure, and feeding 5.3 mg FB1 + FB2/kg induced oxidative damage in the liver at 17 days, and histological damage to the liver and lungs at 21 days, effects on growth were only observed at 21 days [2,44].

Targeted analysis of sphingolipids in heart, gizzard, and breast muscle revealed that the effects of fumonisins varied greatly with the tissue studied. A significant increase in the Sa:So ratio was observed in the heart, and this increase was due to an increase in Sa, with a small increase in So also being observed. The change in sphingoid bases was accompanied by an increase in ceramides and, to a lesser extent, an increase in glycosylceramides and sphingomyelins. Most of the effects observed for dihydroceramides and dihydrosphingomyelins in the heart paralleled those found for ceramides and sphingomyelins. The effect of fumonisins on Sa is consistent with its well-known property of inhibiting CerS (Figure 1), and de novo synthesis of ceramides [10]. An increase in So concomitant with an increase in Sa has previously been reported to be the consequence of sphingomyelin hydrolysis occurring to maintain ceramide concentrations in cells [11], with this mechanism being known as the salvage synthesis pathway of ceramides (Figure 1). In this study, an increase rather than a decrease in sphingomyelins was observed, suggesting that sphingomyelin hydrolysis cannot explain the observed increase in So and ceramides. Moreover, glycosylceramides are derived from ceramides, and because glycosylceramides were increased in this study, the hypothesis of an increase in ceramides in heart secondary to hydrolysis of sphingomyelins appears unlikely (Figure 1). Other mechanisms that could explain an increase in ceramide concentrations in heart include: (1) change in the availability of the substrates used for their de novo synthesis [45,46]; (2) activation of de novo synthesis by different mechanisms linked with an inflammatory response such as Toll-like receptor (TLR) or tumor necrosis factor alpha (TNFα) activation [47]; and (3) increased delivery of sphingolipids from plasma, which includes recycling of sphingolipids from the gut microbiota [46,48].

Irrespective of the mechanism involved, sphingolipid concentrations were increased in the hearts of chickens in this study that had been fed fumonisins. PLS-DA revealed that the changes in sphingolipids were enough to distinguish chickens fed the control diet from chickens fed the fumonisin diet. Analysis of the correlations among sphingolipids also revealed that nearly all the sphingolipids positively correlated together in chickens fed fumonisins, which was not the case in the controls, all suggesting that pronounced changes in sphingolipid homeostasis occurred in the hearts of chickens fed fumonisins. Moreover, it is interesting to note that the increases in sphingolipids for which the fatty acids had 20–26 carbon chain lengths were more pronounced than for those with 14–16 carbon chain lengths. This observation is consistent with previous results in liver and kidney suggesting that fumonisins preferentially inhibit CerS5/6 activity in chickens [13,22]. Thus, it can be hypothesized that a general increase in sphingolipid synthesis occurred in the hearts of animals fed fumonisins, with this increase being less pronounced for sphingolipids for which the fatty acid chain length was 14 or 16 carbons due to partial inhibition in CerS5/6 activity.

Changes in sphingolipid concentrations in the hearts of chickens fed fumonisins could have negative consequences on health. Indeed, recent studies have revealed a prominent role of sphingolipids in cardiovascular diseases and heart failure [31]. Increased concentrations of ceramides in the myocardium have been reported in a rat model of ischemic reperfusion injury [49], and measurement of the expression of serine palmitoyl-transferase in mice suggested an increase in the de novo synthesis of sphingolipids in the infarct area and in the area at risk [50]. By contrast, sphingosine 1-phosphate has been reported to have a protective function in the myocardium [31]. Interestingly, a pronounced increase in So1P was observed in the plasma of chickens fed fumonisins, and a rapid increase in So1P has been reported in transient ischemia and reperfusion injury in humans [13,51]. Unfortunately, So1P was only found at trace levels in the myocardium in this study, and it was hence not possible to quantify its concentration. Only a few studies to date have reported cardiac alterations in animals fed fumonisins. A reduced heart rate has been reported to occur in pigs before the development of pulmonary edema and death [27]. Hypertrophy of the heart has been reported in pigs at very high doses of fumonisins in feed without histopathologic alterations despite an increase in the Sa:So ratio [28]. In contrast, a recent study of low doses of fumonisins revealed no alteration in heart weight, whereas histopathologic lesions characterized by hemorrhage and lymphocyte inflammatory infiltrate were observed [30]. Myocardial infiltration by lymphocytes leading to inflammatory damage is consistent with the hypothesis of activation of de novo synthesis of sphingolipids by TLR or TNFα [30,47]. Cardiac alterations characterized by macroscopic thinning of the heart and thinning of cardiomyocytes have been observed in Japanese quails fed fumonisins [29]. Transmission electron microscopy also revealed that the number of mitochondria was increased and that the mitochondria appeared swollen and pleomorphic in fumonisins-fed quails, although the outer membrane remained intact [29]. Alteration of the heart mitochondria of Japanese quails fed fumonisins is of considerable relevance because an in vitro study has revealed that ceramides cause mitochondrial dysfunction, oxidative stress, and cell death in cardiomyocytes [52]. Interestingly, overexpression of CerS2 induced oxidative stress, mitophagy, and apoptosis, which were prevented by depletion of CerS2. By contrast overexpression of CerS5 did not affect these processes, suggesting a chain-length dependent impact of ceramides on mitochondrial function [52].

A minor decrease in the Sa concentration was observed in the gizzards of chickens fed fumonisins, whereas fumonisins had no effect on the sphingolipid concentrations in breast muscle in this study. There are no data to compare the effects of fumonisins on sphingolipids in gizzard and breast muscle with our present results. Fumonisins have been reported to induce gizzard ulceration in chickens and have been reported to increase the relative weights of gizzards in chickens, turkeys, and ducks, but the mechanism underlying these alterations remains unknown [32,33,34,35,36]. In this study, the decrease in Sa observed in the gizzards of chickens receiving fumonisins was accompanied by a small decrease in So, so the Sa:So remained unchanged. Total ceramides in gizzards were decreased by fumonisins, and all of the ceramides appeared to decrease. The concentrations of dihydroceramides, sphingomyelins, dihydrosphingomyelins, hexosylceramides, and lactosylceramides paralleled those of ceramides. Interestingly, the effects of fumonisins on sphingolipids in gizzards appeared to be independent of the fatty acid carbon chain length. PLS-DA allowed the chickens to be separated into two groups according to the presence or absence of fumonisins in the feed, but the correlations measured among the sphingolipids in gizzards did not differ greatly in fumonisin-fed chickens and in controls. Finally, the effects of fumonisins on gizzards appeared to be consistent with the hypothesis of a decrease in the de novo synthesis of ceramides even though there was no accumulation of Sa. The decrease in sphingolipids in gizzards appeared to be independent of the fatty acid chain, which is different from what was observed in the liver and kidneys and to a lesser extent in the heart [22].

Sphingolipids have been shown to have important roles in skeletal muscle in obesity and aging. Accumulation of C16 ceramide occurs during the development of insulin resistance and CerS6 silencing may be a potential target for the treatment of insulin resistance, obesity, and type 2 diabetes [37]. C18 ceramide, produced mainly by CerS1, also accumulated in mice fed with a high-fat diet that promotes systemic insulin resistance [38,40]. By contrast, a decrease in C16 and C18 ceramides appears to be important in the skeletal muscle and myocardium of aged mice and humans [39]. Histological analysis of muscle in CerS1- and CerS5-deficient mice revealed reduced caliber sizes in slow (type 1) and fast (type 2) fibers of quadriceps femoris [39]. All of these results suggest that the decrease in Cer concentrations observed in gizzards could affect its contractility, even though no clinical consequences were observed.

## 4. Conclusions

In conclusion, this study revealed for the first time that feeding fumonisins at a concentration that did not alter performances in chickens nonetheless resulted in several changes in the sphingolipid composition of muscles that varied according to the type of muscle. Whereas no effects of fumonisins were noted in breast muscle, an increase in sphingoid bases, dihydroceramides, ceramides, and glycosylceramides, and to a lesser extent also sphingomyelins, was observed in the myocardium. These changes were more pronounced for sphingolipids with fatty acid chains of 20 to 26 carbons than for those with 14–16 carbon chain lengths. By contrast, a decrease in dihydroceramides and ceramides, and to a lesser extent also sphingomyelins, was observed in the gizzard. The decreases in sphingolipid levels in the gizzard appeared to be the same irrespective of the fatty acid chain length. Taken together, these results confirmed that the effects of fumonisins on sphingolipids at the scale of the organism are complex and probably involve cell- or metabolite-mediated effects in association with the inhibition of CerS activity. Further studies are necessary to understand these mechanisms and their consequences on health.

## 5. Materials and Methods

### 5.1. Analytes and Reagents

The analytes and reagents used in this study were obtained from Sharlab (Sharlab S.L., Sentmenat, Spain) or Sigma (Sigma Aldrich Chimie, Saint-Quentin-Fallavier, France). The separation of the analytes by UHPLC-MSMS was done with LC-MS grade solvents, whereas all other reagents were HPLC grade. The 42 sphingolipids used as standards were obtained from Sigma or Bertin (Bertin Technologies, Montigny-Le-Bretonneux, France) and corresponded to: deoxysphingosine (dSo = m18:1); deoxysphinganine (dSa = m18:0); sphingosine (So = d18:1); sphinganine (Sa = d18:0); sphingosine-1-P (d18:1P); sphinganine-1-P (d18:0P); glucosylsphingosine (GluSo); lysosphingomyelin (LysoSM); lactosylsphingosine (LacSo); N-acetylsphingosine (18:1/2:0); N-acetylsphinganine (18:0/2:0); ceramides: 18:1/14:0, 18:1/16:0, 18:1/18:0, 18:1/20:0, 18:1/22:0, 18:1/24:1, and 18:1/24:0; ceramide-1P: 18:1/16:0P; dihydroceramides: 18:0/16:0, and 18:0/24:0; deoxyceramides: m18:1/16:0, m18:1/22:0, m18:1/24:1, m18:1/24:0; deoxydihydroceramides: m18:0/22:0, m18:0/24:1, m18;0/24:0; glucosylceramides: Glu18:1/16:0 and Glu18:1/24:1; lactosylceramides: Lac18:1/16:0 and Lac18:1/24:1; ceramides sulfatides: ST18:1/24:1 and ST18:1/24:0; sphingomyelins: SM18:1/14:0, SM18:1/16:0, SM18:1/18:0, SM18:1/18:1, SM18:1/20:0, SM18:1/22:0, SM18:1/24:1, and SM18:1/24:0. The 12 sphingolipids used as IS corresponded to the 10 IS mixture “Ceramide/Sphingoid Internal Standard Mixture I” from Avanti Polar Lipids, which contains 25 µM C17 sphingosine (d17:1), C17 sphinganine (d17:0), C17 sphingosine-1-P (d17:1P), C17 sphinganine-1-P (d17:0P), lactosyl (ß) C12 ceramide (Lac18:1/12:0), C12 sphingomyelin (SM18:1/12:0), glucosyl (ß) C12 ceramide (Glu18:1/12:0), 12:0 ceramide (18:1/12:0), 12:0 ceramide-1-P (18:1/12:0P), and 25:0 ceramide (18:1/25:0) in ethanol solution. This mixture was completed by C12 deoxyceramide (m18:1/12:0) and C12 ceramide sulfatide (ST18:1/12:0) solubilized in ethanol at a concentration of 25 µM.

### 5.2. Tissue Samples

Heart, gizzard, and breast muscle were obtained from a Ross chicken study for which the animal maintenance conditions, feed formulation, and results of the effects of fumonisins on health and performance were detailed in [43]. This study was completed at Cebiphar (Cebiphar, ondettes, France) under number V9152 as a randomized, parallel, monocenter study under project number 2017062111426641 accepted by the French Ministry of Higher Education, Research, and Innovation (Paris, France) on 6 November 2017. Briefly, the experimental diets were formulated on a corn-soybean basis to best meet the nutritional needs of the chickens. Corn containing fumonisins was incorporated to a final concentration of FB1, FB2, and FB3 in the feed of 15.1, 5.6, and 0.9 mg/kg, respectively. Mycotoxin-free corn was used as the control diet. Mycotoxins in the raw materials and in the diets were measured by LC-MSMS according to AFNOR V03-110 [53]. Drinking water and feed were provided ad libitum. A diet containing fumonisins was provided to 10 chickens per group from the day of age 14 to the day of age 21. A control diet free of mycotoxins was provided to 10 other chickens until the day of age 21. On day 21, the feed was removed for eight hours before euthanasia and tissue collection. Heart, gizzard, and breast muscle were stored at −80 °C until analysis.

### 5.3. Sphingolipids in Tissues

The sphingolipids in heart, gizzard, and breast muscle were measured as previously described in [13] and completed in [22] for the determination of deoxyceramides, deoxydihydroceramides, and ceramides sulfatides. Briefly, 0.5 g of tissue was homogenized with a Potter grinder in 1.5 mL of phosphate buffer (0.1 M, pH 7.4) and centrifuged for 15 min at 3000× *g*. A 40 µL aliquot of the supernatant was collected and 120 µL of NaCl 0.9%, 600 µL of methanol/chloroform (2:1), and 10 µL of a solution containing the IS mixture were added to obtain a final concentration of each IS equivalent to 6250 pmol/g of tissue. The IS mixture was composed of the “Ceramide/Sphingoid Internal Standard Mixture I” that was completed with m18:1/12:0 and ST18:1/12:0. Samples were incubated overnight at 48 °C. After cooling to room temperature, 100 µL of KOH (1 M in methanol) was added and the samples were incubated for 2 h at 37 °C to hydrolyze glycerophospholipids, which could otherwise interfere with the determination of sphingolipids. A 10 µL aliquot of 50% acetic acid was added and the samples were centrifuged for 15 min at 4500× *g*. The supernatant was collected, and the residue was extracted again with 600 µL of methanol/chloroform (2:1). The second supernatant was added to the first and then evaporated to dryness. The dry residue was solubilized in 200 µL methanol, and a 5 µL aliquot was injected into the UHPLC-MSMS system comprising a 1260 binary pump, an autosampler, and an Agilent 6410 triple quadrupole spectrometer (Agilent, Santa Clara, CA, USA). The analytes were separated on a Poroshell 120 column (3.0 × 50 mm, 2.7 µm). The mobile phase was (A) methanol/acetonitrile/isopropanol (4/1/1) and (B) water, each containing 10 mM ammonium acetate and 0.2% formic acid. The mobile phase was delivered using a gradient of elution at a flow rate of 0.3 mL/min, as previously described [13]. Sphingolipids were detected after positive electrospray ionization under the following conditions: temperature 300 °C, flow rate of 10 L/min, pressure of 25 psi, capillary voltage 4000 V. Transitions, fragmentor voltages, and collision energies were reported in [22]. Agilent Mass-Hunter quantitative analysis software (B.05.00 SP03/Build 5.0.291.7) was used to analyze the chromatograms. The precision of the method expressed as a relative standard deviation (RSD) was considered acceptable for an RSD of 20%. As shown in Appendix A, the linearity of the method of analysis was good over a relatively large range of concentrations, in agreement with previous results [13,22]. The recovery of the 12 IS in heart, gizzard, and breast muscle is presented in Appendix A. The recovery varied according to the tissue and the analyte. The lowest recoveries of the IS were measured in breast muscle, whereas the recoveries in heart and gizzard were similar. A high recovery attributed to a positive matrix effect was observed for 18:1/12P and to a lesser extent for d17:1P and d17:0P, which is in agreement with previous results involving other tissues [13,22]. The lowest recovery was measured for 18:1:25:0, which is in agreement with previous results involving other tissues [13,22]. The repeatability of the method was considered good, with an RSD of 20%. Good repeatability was observed for all the IS in heart, gizzard, and breast muscle except for 18:1/25:0 in gizzard, for which an RSD of 25% was found. This value was considered acceptable, with a high RSD being already found for this analyte in other tissues [13,22].

The sphingolipid concentrations were determined from the calibration curves obtained with the standards. The final concentrations in tissues were corrected by the recovery measured for the corresponding IS. No correction was used for 18:1/2:0, 18:0/2:0, GluSo, LysoSM, and LacSo. The concentrations of sphingolipids not available as standards were calculated using the calibration curves obtained with standards of the same class with the closest mass and similar abundance. The final concentrations in tissues were corrected as carried out for the sphingolipids available as standards.

### 5.4. Analytes and Reagents

One-way ANOVA was carried out to compare the sphingolipid concentrations in the controls and in the chickens fed fumonisins after determination of the homogeneity of variance (Hartley’s test). Significant differences were reported as follows: (*) 0.05 < *p* < 0.01, (**) 0.01 < *p* < 0.001, and (***) *p* < 0.001. Partial least squares-discriminant analysis (PLS-DA) was carried out to reveal the appurtenance of the chickens to the control and the fumonisins-fed group and to identify the sphingolipids that were important in the model. A score above 1.1 was retained to select the variables important in the projection (VIP) for heart and gizzard, whereas a score above 0.8 was retained for breast muscle. Pearson coefficients among the sphingolipids assayed in this study were measured and reported in bold font when significant in Appendix A. All the statistical analyses were carried out with XLSTAT Biomed software (Addinsoft, Bordeaux, France).

## Figures and Tables

**Figure 2 toxins-14-00828-f002:**
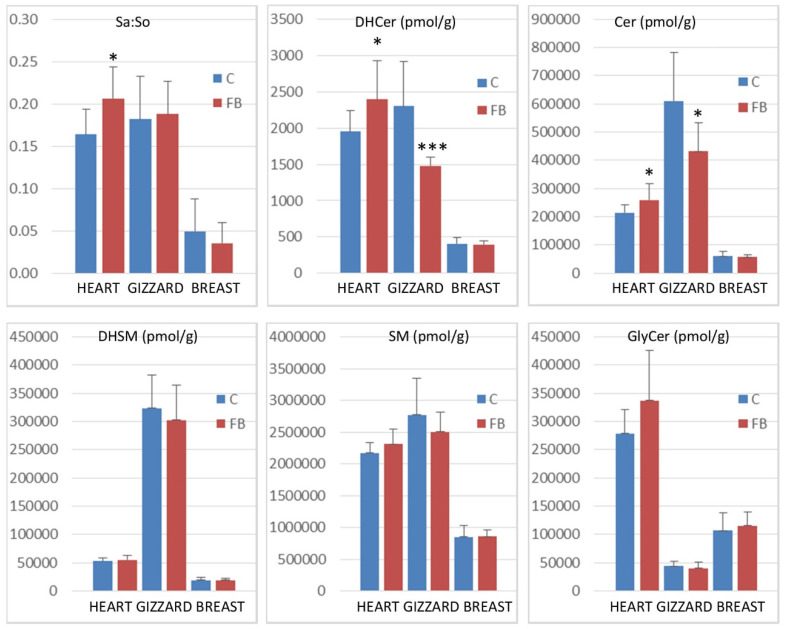
Sphinganine:sphingosine (Sa:So) ratios and concentrations of sphingolipid reported as the total of dihydroceramides (DHCer), ceramides (Cer), dihydrosphingomyelins (DHSM), sphingomyelins (SM), and glycosylceramides (GlyCer) in the heart, gizzard, and breast muscle of chickens fed the control diet free of mycotoxins (C) and chickens fed 9 days with a diet containing 20.8 mg FB1 + FB2/kg (FB). * 0.05< *p* <0.01 and *** *p* < 0.001.

**Figure 3 toxins-14-00828-f003:**
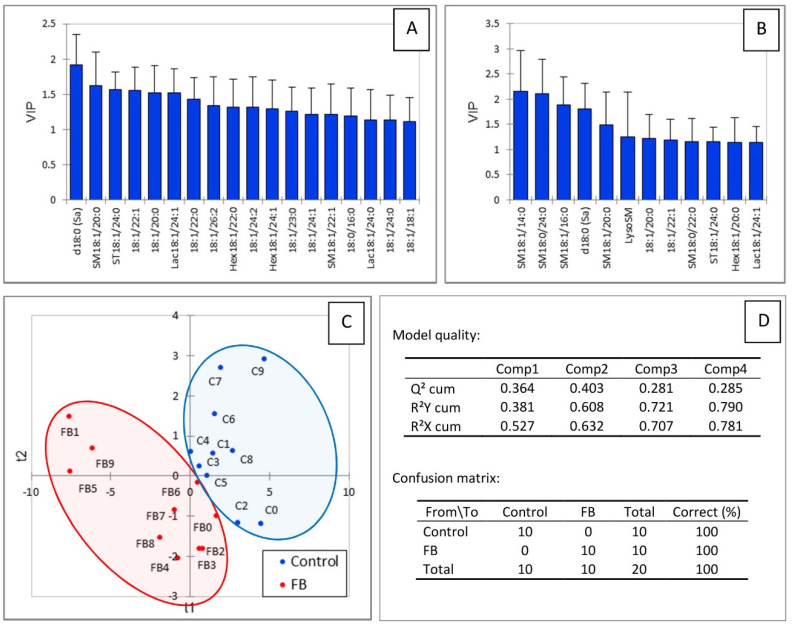
PLS-DA of sphingolipids measured in the heart of chickens fed 9 days with a control diet free of mycotoxins or a diet containing 20.8 mg FB1 + FB2/kg. (**A**) Scores of the VIP for the first component, and (**B**) the second component. (**C**) Discrimination on the factor axes extracted from the original explanatory variables. (**D**) Quality of the model and confusion matrix for the training sample (variable groups).

**Figure 4 toxins-14-00828-f004:**
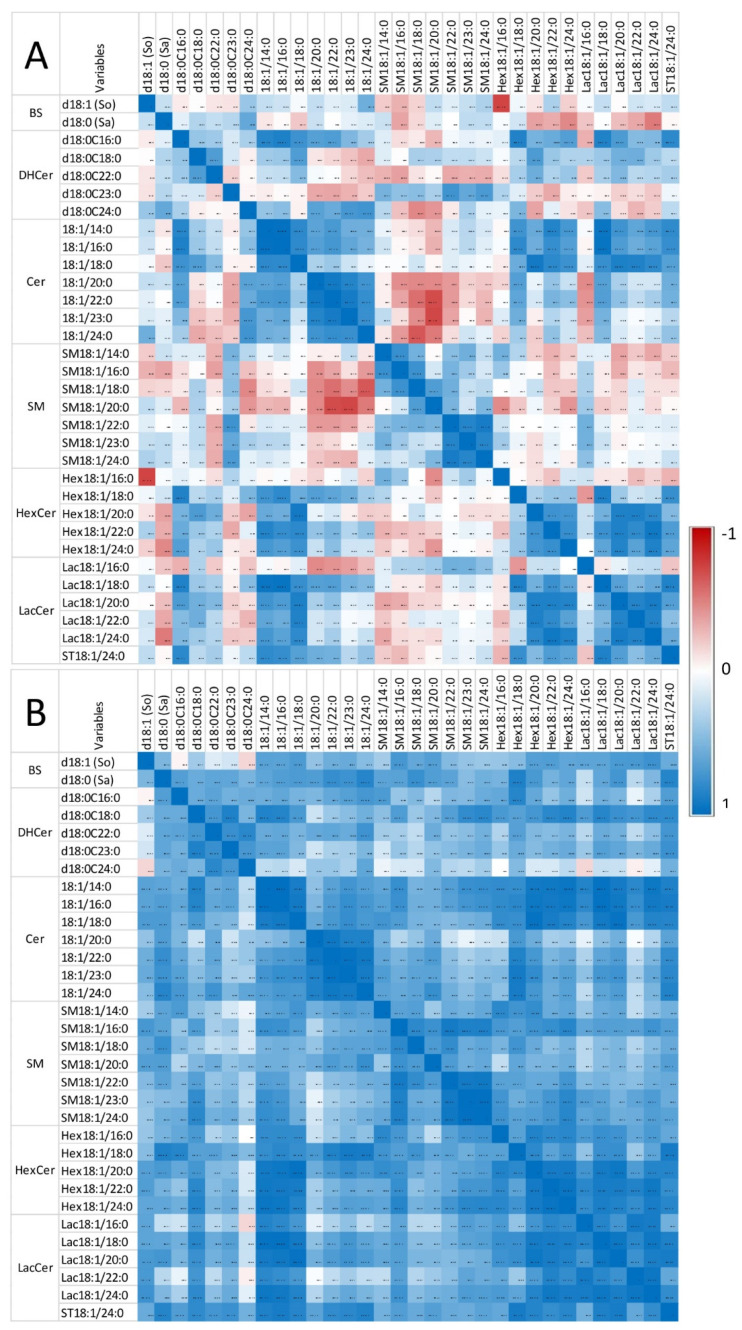
Correlation heatmap of So, Sa, and sphingolipids with 14 to 24 carbon fatty acid saturated chain lengths observed in the heart of chickens fed the control diet (**A**) and chickens fed 9 days with a diet containing 20.8 mg FB1 + FB2/kg (**B**). Numeric values of the Pearson coefficients of correlation observed among all sphingolipids assayed in this study are reported in Appendix A.

**Figure 5 toxins-14-00828-f005:**
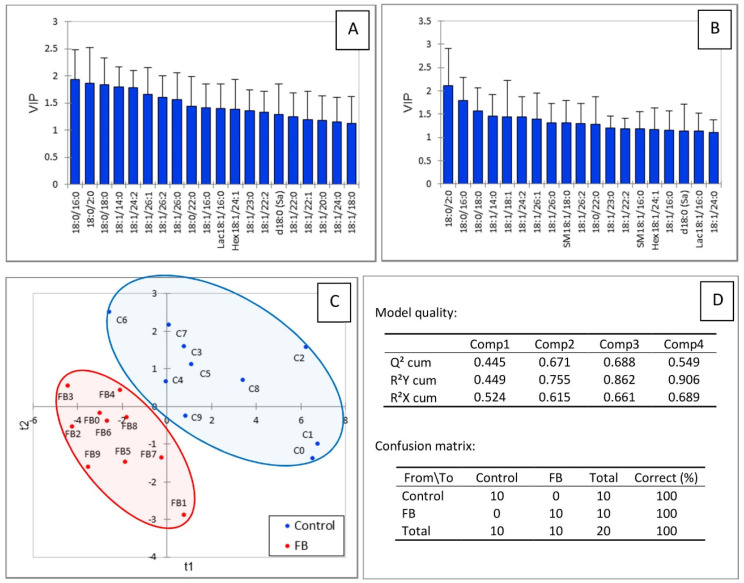
PLS-DA of sphingolipids measured in the gizzards of chickens fed for 9 days with a control diet free of mycotoxins or a diet containing 20.8 mg FB1 + FB2/kg. (**A**) Scores of the VIP for the first component and (**B**) the second component. (**C**) Discrimination on the factor axes extracted from the original explanatory variables. (**D**) Quality of the model and confusion matrix for the training sample (variable groups).

**Figure 6 toxins-14-00828-f006:**
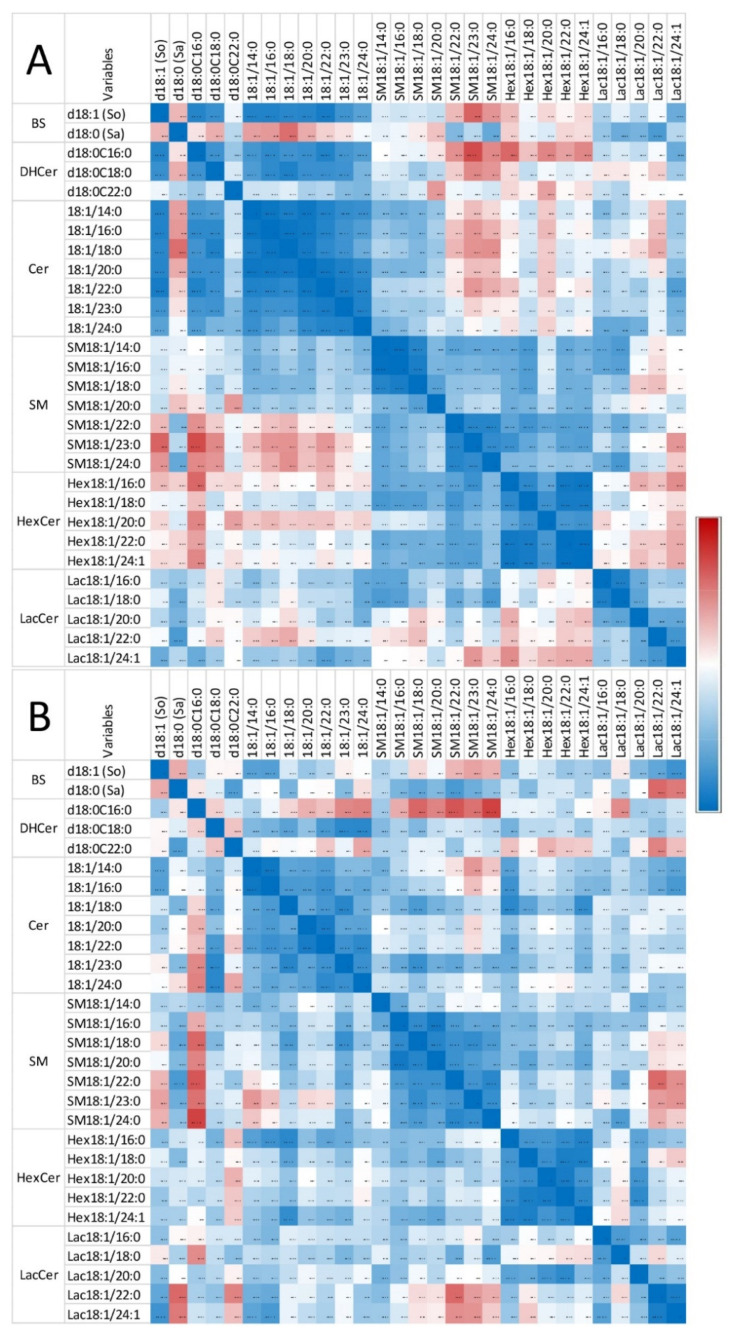
Correlation heatmap of So, Sa, and sphingolipids with 14 to 24 carbon fatty acid saturated chain lengths observed in the gizzard of chickens fed the control diet (**A**) and chickens fed 9 days with a diet containing 20.8 mg FB1 + FB2/kg (**B**). Numeric values of the Pearson coefficients of correlation observed among all sphingolipids assayed in this study are reported in Appendix A.

**Table 1 toxins-14-00828-t001:** Concentrations of sphingolipids measured in the heart, gizzard, and breast muscle of broilers fed a control diet free of mycotoxins and broilers fed 20.8 mg FB1 + FB2 for 9 days ^1^.

	Heart		Gizzard		Breast Muscle	
	Control	FB	Control	FB	Control	FB
Sphingoid bases and derivates						
d18:1 (So)	5326 ± 596	5891 ± 1086	1367 ± 387	1157 ± 183	2585 ± 636	3107 ± 388 *
d18:0 (Sa)	868 ± 133	1204 ± 222 ***	233 ± 19	213 ± 24 *	127 ± 89	110 ± 72
18:1/2:0	62 ± 13	69 ± 11	27 ± 8	29 ± 9	24 ± 8	23 ± 7
18:0/2:0	ND	ND	16 ± 5	25 ± 7 **	ND	ND
LysoSM	249 ± 41	234 ± 33	353 ± 34	327 ± 47	130 ± 29	122 ± 23
Ceramides and dehydroceramides						
18:1/14:0	5277 ± 894	6217 ± 1588	307 ± 85	204 ± 45 **	75 ± 22	70 ± 9
18:1/16:0	77,884 ± 12,241	90,785 ± 21,079	110,479 ± 33,863	81,436 ± 18,040 *	14,530 ± 5548	13,523 ± 2337
18:0/16:0	1125 ± 193	1353 ± 298	1753 ± 533	1089 ± 104 **	375 ± 79	355 ± 40
18:1/18:1	1036 ± 225	1232 ± 243	343 ± 132	329 ± 102	ND	ND
18:1/18:0	25,264 ± 4163	30,555 ± 10,317	22,600 ± 8108	17,194 ± 4879	15,646 ± 4753	15,281 ± 3584
18:0/18:0	62 ± 20	82 ± 35	328 ± 81	225 ± 44 **	33 ± 12	40 ± 13
18:1/20:0	17,052 ± 2930	22,897 ± 5917 *	17,282 ± 5215	13,521 ± 3411	3574 ± 1426	3510 ± 889
18:1/22:2	1556 ± 344	1882 ± 549	49,559 ± 14,820	36,320 ± 11,872 *	308 ± 97	290 ± 67
18:1/22:1	681 ± 130	972 ± 292 *	12,002 ± 5214	8371 ± 2826	314 ± 103	299 ± 59
18:1/22:0	31,576 ± 4466	39,187 ± 8219 *	26,966 ± 7331	21,517 ± 4277	6466 ± 2186	6222 ± 1146
18:0/22:0	60 ± 19	76 ± 33	223 ± 53	168 ± 46 **	ND	ND
18:1/23:1	ND	ND	991 ± 344	772 ± 215	ND	ND
18:1/23:0	2140 ± 341	2553 ± 496 *	7308 ± 1401	6070 ± 988 *	445 ± 128	402 ± 80
18:0/23:0	23 ± 9	31 ± 12	ND	ND	ND	ND
18:1/24:2	3273 ± 714	4170 ± 1002 *	272,361 ± 78,578	177,125 ± 43,980 **	1380 ± 435	1377 ± 292
18:1/24:1	25,974 ± 4601	32,914 ± 9442	71,565 ± 28,535	54,187 ± 16,562	11,736 ± 3967	11,664 ± 1939
18:1/24:0	20,773 ± 2780	23,963 ± 4458	14,143 ± 2922	12,127 ± 1801	4563 ± 1261	4399 ± 1009
18:0/24:0	683 ± 139	852 ± 349	ND	ND	ND	ND
18:1/26:2	539 ± 125	713 ± 197 *	2815 ± 985	1862 ± 383 *	26 ± 20	21 ± 8
18:1/26:1	444 ± 91	526 ± 135	1270 ± 333	881 ± 243 *	32 ± 16	33 ± 17
18:1/26:0	ND	ND	230 ± 82	151 ± 38 *	ND	ND
Sphingomyelins and dehydrosphingomyelins						
SM18:1/14:0	13,516 ± 2450	12,256 ± 1114	1962 ± 424	1749 ± 274	1024 ± 250	1054 ± 150
SM18:1/16:0	279,024 ± 24,417	269,938 ± 20,805	296,493 ± 37,427	283,011 ± 26,615	142,118 ± 28,443	134,059 ± 11,388
SM18:0/16:0	29,224 ± 3338	30,528 ± 3387	287,685 ± 53,159	268,997 ± 52,074	14,195 ± 4236	14,297 ± 2589
SM18:1/18:1	7638 ± 1747	8262 ± 1449	ND	ND	ND	ND
SM18:1/18:0	718,620 ± 70,463	774,687 ± 75,203	455,089 ± 63,676	452,100 ± 59,844	557,238 ± 118,116	576,469 ± 66,525
SM18:0/18:0	9964 ± 2069	10,078 ± 2031	20,824 ± 4824	19,992 ± 6472	2876 ± 682	2993 ± 579
SM18:1/20:0	131,232 ± 8241	147,199 ± 14,149 **	46,568 ± 5174	46,010 ± 6491	33,633 ± 10,149	34,817 ± 5905
SM18:0/20:0	1780 ± 228	1898 ± 404	2872 ± 646	2899 ± 819	413 ± 94	414 ± 131
SM18:1/22:2	7332 ± 1102	8022 ± 786	70,886 ± 24,225	61,750 ± 14,671	2554 ± 559	2480 ± 283
SM18:1/22:1	8259 ± 747	9199 ± 1216	32,164 ± 5132	31,112 ± 8408	4092 ± 1083	3931 ± 687
SM18:1/22:0	509,447 ± 41,889	546,720 ± 65,977	221,170 ± 32,543	201,609 ± 30,819	69,740 ± 18,958	69,604 ± 15,227
SM18:0/22:0	5074 ± 925	5230 ± 1059	3921 ± 1054	3492 ± 1033	520 ± 140	498 ± 102
SM18:1/23:1	7885 ± 1113	8422 ± 1406	5465 ± 1250	4849 ± 958	2787 ± 927	2969 ± 955
SM18:1/23:0	27,322 ± 3979	29,310 ± 5566	33,713 ± 7600	34,101 ± 12,246	2134 ± 578	2062 ± 468
SM18:0/23:0	3984 ± 755	4392 ± 1105	466 ± 90	423 ± 115	765 ± 269	705 ± 180
SM18:1/24:3	ND	ND	19 ± 2	18 ± 1	23 ± 1	24 ± 2
SM18:1/24:2	30,205 ± 4115	31,127 ± 3310	1078,170 ± 385,433	876,991 ± 143,010	14,609 ± 4097	14,083 ± 2864
SM18:1/24:1	254,344 ± 30,783	271,904 ± 47,893	381,839 ± 65,468	378,196 ± 73,027	57,450 ± 14,974	58,443 ± 14,194
SM18:1/24:0	152,411 ± 18,191	166,908 ± 26,357	76,543 ± 12,027	70,781 ± 14,501	15,543 ± 4195	15,449 ± 3559
SM18:0/24:1	2453 ± 434	2665 ± 483	6900 ± 2267	6308 ± 2712	319 ± 75	333 ± 89
SM18:0/24:0	605 ± 104	564 ± 124	954 ± 281	794 ± 179	ND	ND
SM18:1/25:2	3295 ± 2038	3682 ± 2300	7145 ± 1906	6169 ± 1291	239 ± 60	211 ± 38
SM18:1/25:1	1527 ± 243	1771 ± 348	5960 ± 1313	5792 ± 1904	303 ± 77	285 ± 93
SM18:1/25:0	817 ± 215	948 ± 221	1493 ± 422	1478 ± 453	108 ± 26	106 ± 28
SM18:1/26:3	5393 ± 677	5784 ± 586	6079 ± 2098	5165 ± 1483	324 ± 119	381 ± 123
SM18:1/26:2	8088 ± 1363	9027 ± 1297	35,708 ± 11,840	30,033 ± 8733	529 ± 182	565 ± 217
SM18:1/26:1	2500 ± 483	2742 ± 485	12,702 ± 3216	11,639 ± 3881	307 ± 101	329 ± 97
SM18:1/26:0	ND	ND	857 ± 245	841 ± 291	ND	ND
Hexosyl-, and lactosylceramides, and ceramides sulfatides						
Hex18:1/16:0	2695 ± 477	2789 ± 640	730 ± 250	601 ± 143	832 ± 257	781 ± 203
Hex18:1/18:0	20,024 ± 4349	23,297 ± 5049	2913 ± 1341	2329 ± 1234	ND	ND
Hex18:1/20:0	3123 ± 538	3534 ± 1098	676 ± 283	688 ± 419	ND	ND
Hex18:1/22:0	2199 ± 370	2811 ± 752 *	1179 ± 716	874 ± 385	ND	ND
Hex18:1/24:1	14,817 ± 1554	17,483 ± 3396 *	1160 ± 660	637 ± 261 *	1340 ± 976	976 ± 518
Hex18:1/24:0	4285 ± 875	5119 ± 1405	934 ± 351	905 ± 466	ND	ND
Lac18:1/16:0	3280 ± 663	3293 ± 1192	5131 ± 1257	3919 ± 1036 *	1172 ± 441	1091 ± 162
Lac18:1/18:0	164,969 ± 26,340	195,360 ± 49,026	24,678 ± 6030	23,262 ± 8744	16,877 ± 6682	18,010 ± 4202
Lac18:1/20:0	16,080 ± 3467	20,537 ± 7002	971 ± 667	764 ± 424	7237 ± 3232	7812 ± 2430
Lac18:1/22:0	11,757 ± 4980	17,071 ± 10,225	2325 ± 821	2803 ± 1798	1432 ± 654	1783 ± 610
Lac18:1/24:1	17,053 ± 2330	22,119 ± 5235 *	3281 ± 970	3747 ± 2019	18,040 ± 8822	23,515 ± 7061
Lac18:1/24:0	18,598 ± 3893	24,212 ± 8371	ND	ND	2308 ± 1149	2441 ± 785
ST18:1/24:1	39,266 ± 6545	46,422 ± 11,226	ND	ND	ND	ND
ST18:1/24:0	56,643 ± 9617	71,979 ± 13,510 **	ND	ND	ND	ND

^1^ Results are expressed in pmol sphingolipids/g as means ± SD, *n* = 10. ANOVA was used to assess the difference between groups. Significant differences among groups were reported as follow: (*) 0.05 < *p* <0.01; (**) *p* < 0.01; (***) *p* < 0.001. ND: not detected.

## Data Availability

Not applicable.

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
