# Peer review of "Targeted Sphingolipid Analysis in Heart, Gizzard, and Breast Muscle in Chickens Reveals Possible New Target Organs of Fumonisins"

_toxins, 2022, doi:10.3390/toxins14120828_

Round 1
Reviewer 1 Report
My comments can be found in the attached MS.

Author Response
Thank you for your comments and the time you took to read our manuscript. All the corrections suggested have been incorporated in the revised version. Below are the answers to the comments/questions
Abstract is a bit lengthy.
Abstract was shortened as suggested
L36: Isn't it a contaminant? If the crop is not infected with Fusarium then the fodder is free of Fumonisins.
I am not sure to have well understood, the word “compound” was replaced by “contaminant”
L79, L83: Rewrite this one to avoid the monotony.
Sentence was rewritten as suggested
L95: Curious to know why 9 days? Were chickens (treated) sick beyond those days?
The farms are supplied with feed every 5 to 9 days. It is unlikely that animals will be exposed twice in a row to a batch contaminated to its maximum value. 9 days therefore seems a realistic exposure time, without constituting a worst case situation. Much higher levels are required to show an effect on performance
L368: In which host? Pigs or chickens?
In pigs, this point was clarified
L375: This one was mentioned in Introduction.
The reference is cited in the introduction but we did not want to overload this part of the manuscript by detailing the observed effects.
L393: in treated chickens, rewrite this sentence for clarification.
sentence was rewritten to avoid conclusion
L517 : Please mention the instrument used.
This information was given L493-494: "UHPLC-MSMS system comprising a 1260 binary pump, an autosampler, and an Agilent 6410 triple quadrupole spectrometer (Agilent, Santa Clara, CA, USA).
Reviewer 2 Report
The paper describes the LC-MS/MS based sphingolipidome analysis in some organ tissue of chickens and investigation of the effect of fumonisins, FB1 and FB2.
The analytical method for sphingolipidome analysis was designed by many authentic standards as wide targeted metabolome analysis and so it was considered highly accurate method.
I have a few comments.
1) Tables and figures; Unit was not shown in both figures and tables. They should be expressed.
2) Figure 4 and 6; I think that colors of red and blue might be replaced each other.
3) Conclusions; Summarized graphical abstract should be added in conclusions.
Author Response
The paper describes the LC-MS/MS based sphingolipidome analysis in some organ tissue of chickens and investigation of the effect of fumonisins, FB1 and FB2.
The analytical method for sphingolipidome analysis was designed by many authentic standards as wide targeted metabolome analysis and so it was considered highly accurate method.
Thank you for your positive feedback on this manuscript and the time you took to read it
I have a few comments.
1) Tables and figures; Unit was not shown in both figures and tables. They should be expressed.
Thanks for the comment, the captions of the Figures and Tables have been reviewed but I don't see what can be completed.
Table 1: foot note says: “Results are expressed in pmol sphingolipids/g”
Figure 2: units are specified except for the ratio Sa:So which being a ratio is without units.
Figure 3 and 5: the VIPs are unitless, they are proportional values created by the modelling, they do not correspond to concentrations. The same is true for the projections on the t1 and t2 axes and the model parameters.
Figure 4 and 6: these are correlations, therefore without units, the numerical values are given in Tables S1 and S2.
We are of course ready to correct this point if we have missed any additional information.
2) Figure 4 and 6; I think that colors of red and blue might be replaced each other.
Thanks for the comment, by convention, colour used in genetics for heatmap are: red = negative, green = positive. To avoid confusion we have replaced green by blue, but it seems appropriate to keep red for negative correlations. We are of course ready to correct this point if we have missed any additional information.
3) Conclusions; Summarized graphical abstract should be added in conclusions.
Thanks for the comment, a graphical abstract was added to the manuscript